# High-Throughput Screening of Antiviral Compounds Using a Recombinant Hepatitis B Virus and Identification of a Possible Infection Inhibitor, Skimmianine

**DOI:** 10.3390/v16081346

**Published:** 2024-08-22

**Authors:** Mika Yoshita, Masaya Funaki, Tetsuro Shimakami, Masaki Kakuya, Kazuhisa Murai, Saiho Sugimoto, Shotaro Kawase, Koji Matsumori, Taro Kawane, Tomoki Nishikawa, Asuka Nakamura, Reo Suzuki, Atsuya Ishida, Narumi Kawasaki, Yuga Sato, Ying-Yi Li, Ariunaa Sumiyadorj, Kouki Nio, Hajime Takatori, Kazunori Kawaguchi, Kazuyuki Kuroki, Takanobu Kato, Masao Honda, Taro Yamashita

**Affiliations:** 1Department of Gastroenterology, Graduate School of Medicine, Kanazawa University, 13-1 Takaramachi, Kanazawa 920-8641, Japan; muharu1125@gmail.com (M.Y.); funachan1@gmail.com (M.F.); m.kakuya.39@gmail.com (M.K.); yumeoi0812@yahoo.co.jp (S.S.); skawase.0806@gmail.com (S.K.); matsumorikoji@gmail.com (K.M.); barutoest57@gmail.com (T.K.); rbgqy690@gmail.com (T.N.); liyingyi@staff.kanazawa-u.ac.jp (Y.-Y.L.); ariunasumi1972@gmail.com (A.S.); nio@m-kanazawa.jp (K.N.); hajimetakatori@gmail.com (H.T.); kawaguchi@m-kanazawa.jp (K.K.); kkuroki27@gmail.com (K.K.); taroy62m@staff.kanazawa-u.ac.jp (T.Y.); 2Department of Clinical Laboratory Medicine, Graduate School of Medical Science, Kanazawa University, 5-11-80 Kodatsuno, Kanazawa 920-0942, Japan; k.murai.0612@gmail.com (K.M.); asu317tama@stu.kanazawa-u.ac.jp (A.N.); reonandesu@gmail.com (R.S.); atsuya.98617@gmail.com (A.I.); lux67ufrv@stu.kanazawa-u.ac.jp (N.K.); ysa2601@stu.kanazawa-u.ac.jp (Y.S.); mhonda@m-kanazawa.jp (M.H.); 3Department of Virology II, National Institute of Infectious Diseases, 1-23-1 Toyama, Shinjuku-ku, Tokyo 162-8640, Japan; takato@niid.go.jp

**Keywords:** hepatitis B virus, skimmianine, high-throughput screening

## Abstract

We developed a novel hepatitis B virus (HBV) infection-monitoring system using a luminescent, 11-amino acid reporter (HiBiT). We performed high-throughput antiviral screening using this system to identify anti-HBV compounds. After the infection of primary human hepatocytes with the recombinant virus HiBiT-HBV, which contains HiBiT at its preS1, 1262 compounds were tested in a first screening using extracellular HiBiT activity as an indicator of viral infection. Following a second screening, we focused on the compound skimmianine, which showed a potent antiviral effect. When skimmianine was added at the same time as HiBiT-HBV infection, skimmianine inhibited HiBiT activity with EC_50_ of 0.36 pM, CC_50_ of 1.67 μM and a selectivity index (CC_50_:EC_50_ ratio) of 5,100,000. When skimmianine was added 72 h after HiBiT-HBV infection, the EC_50_, CC_50_ and selectivity index were 0.19 μM, 1.87 μM and 8.79, respectively. Time-lapse fluorescence imaging analysis using another recombinant virus, ReAsH-TC155HBV, with the insertion of tetra-cysteine within viral capsid, revealed that skimmianine inhibited the accumulation of the capsid into hepatocytes. Furthermore, skimmianine did not inhibit either attachment or internalization. These results imply that skimmianine inhibits the retrograde trafficking of the virus after internalization. This study demonstrates the usefulness of the recombinant virus, HiBiT-HBV, for high-throughput screening to identify anti-HBV compounds.

## 1. Introduction

Approximately 290 million people worldwide are infected with hepatitis B virus (HBV) [1]. Antiviral therapies comprising nucleos(t)ide analogs and/or interferon (IFN) are used in active chronic HBV carriers. These therapies can delay the progression of HBV-related diseases such as liver cirrhosis and hepatocellular carcinoma but cannot completely eliminate HBV due to the persistence of HBV covalently closed circular DNA (cccDNA) in hepatocytes. Therefore, novel antiviral agents that can eliminate cccDNA from HBV-infected hepatocytes are urgently needed to cure HBV infection. For the development of such agents, efficient HBV cell culture systems are needed that mimic the entire life cycle of HBV and can be used to easily monitor HBV infection and/or replication.

To achieve such a goal, the ideal approach is to create reporter- or marker-expressing recombinant HBV. Such tools will make it much easier to measure the levels of reporters or markers as indicators of HBV infection and/or replication compared with the labor-intensive measurement of the amount of HBV genomes or proteins. However, it was thought to be difficult to insert a foreign gene into the HBV genome and produce recombinant HBV without impairing viral fitness due to the complexity of the HBV genome structure and replication process [2].

To overcome the difficulties associated with inserting a foreign gene into the HBV genome without severely affecting its life cycle, we applied nano-luciferase (NanoLuc) binary technology (NanoBiT) [3]. NanoBiT is a split reporter comprising two subunits: high-affinity NanoBiT (HiBiT) and large NanoBiT [4]. The individual subunits do not possess enzymatic activity but the complex regains its NanoLuc enzymatic activity when HiBiT and large NanoBiT associate. HiBiT comprises 11 amino acids (VSGWRLFKKIS), and we expected HiBiT insertion into the HBV genome to minimally reduce HBV fitness due to its small size.

In previous studies, we inserted the HiBiT coding sequence into the N-terminus of preS1 in a 1.2-fold plasmid encoding a genotype C HBV genome and then prepared a recombinant cell culture-derived virus (HiBiT-HBVcc) after the transfection of this HiBiT-containing plasmid into a human hepatoblastoma cell line, HepG2. After primary human hepatocytes (PXB cells) were infected with the HiBiT-HBVcc, HiBiT-HBV was able to replicate, produce infectious virus and re-infect PXB cells, even though the synthesis of relaxed-circular DNA from single-stranded DNA in HiBiT-HBV was decreased to one-third of that of wild-type HBV and the infectivity of HiBiT-HBVcc was decreased to one-tenth of that of wild-type HBVcc. Notably, the entire life cycle of HBV can be monitored by the measurement of extracellular HiBiT activity [5]. In this study, we performed high-throughput antiviral screening of two kinds of compound libraries using HiBiT-HBVcc infection/replication in PXB cells. Finally, we identified a furoquinoline alkaloid, skimmianine, as a compound showing the inhibitory effect of HBV infection.

## 2. Materials and Methods

### 2.1. Plasmids

pUC1.2×HBV-C/HiBiT [5] plasmid encoding a 1.2-fold HBV genome (isolate C_JPNAT, genotype C, accession number AB246345 [6]) with the insertion of the HiBiT coding sequence into the N-terminus of preS1 was used to prepare the cell culture-derived recombinant virus HiBiT-HBV(Gt-C). The HiBiT coding sequence was newly inserted into the C-terminus of the NTCP binding site of preS1 in a HBV genome (subtype ayw, genotype D, accession number U95551) within the plasmid of which HBV pregenomic RNA transcription is regulated by a CMV promoter. This plasmid, pCMV HBV-D/HiBiT, was used to prepare the cell culture-derived recombinant virus HiBiT-HBV(Gt-D). pUC1.2×HBV-C/TC155 [7] plasmid, encoding a 1.2-fold HBV genome (isolate C_JPNAT, genotype C, accession number AB246345 [6]) with the insertion of tetracysteine (C-C-P-G-C-C) at the 155th capsid amino acid, was used to prepare ReAsH-TC155HBV [7].

### 2.2. Cells

HepG2-NTCP-C4 cells are a subline of a human hepatoblastoma cell line HepG2 that stably express NTCP [8], and HepG2.2.15 cells are also a subclone of HepG2, which supports stable HBV replication [9]. Both HepG2-NTCP-C4 cells and HepG2.2.15 cells were obtained from Dr Koichi Watashi (National Institute of Infectious Diseases, Tokyo, Japan). HepAD38 cells are a subclone of HepG2 and support HBV replication under conditions that can be regulated with tetracycline [10], and this cell line was obtained from Dr. Christoph Seeger (Fox Chase Cancer Center, Philadelphia, PA, USA). HepG2-NTCP-YFP cells were a subclone of HepG2 that stably express the NTCP-YFP fusion protein, as was established in our own laboratory. These cells were cultured in Dulbecco’s modified Eagle’s medium (Thermo Fisher Scientific, Waltham, MA, USA) supplemented with 10% fetal bovine serum (Thermo Fisher Scientific), 1% L-glutamine (Thermo Fisher Scientific) and 1% penicillin/streptomycin (Thermo Fisher Scientific) in a humidified atmosphere of 5% CO_2_ at 37 °C. PXB cells, which are primary human hepatocytes, were isolated from chimeric PXB mice with humanized livers, and the purity of the isolated human hepatocytes exceeded 90% [11,12]. PXB cells were purchased from and provided by Phoenix Bio (Hiroshima, Japan), seeded in 96-well plates at a density of 2.1 × 10^5^ cells/cm^2^ and cultured according to the manufacturer’s instructions.

### 2.3. Virus Preparations and Infection

HepG2-NTCP-C4 cells were transfected with the plasmids encoding each recombinant virus using Lipofectamine 3000 (Thermo Fisher Scientific). The medium was harvested 3 and 7 days after transfection and was concentrated about 30-fold by centrifugation with Amicon Ultra centrifugal filter units (100 kDa) (Merck Millipore, Burlington, MA, USA). ReAsh-TC155HBV was labeled with ReAsH fluorescent dye [7]. PXB cells or HepG2-NTCP-C4 cells were inoculated with recombinant HBVcc for infection at 50,000 genome equivalents (gEq)/cell in the presence of 4% PEG 8000 (Merck Millipore) and 2% DMSO overnight.

Genotype C HBV, which was derived from the sera of PXB mice infected with HBV, was purchased from Phoenix Bio and used to infect PXB cells at 5 gEq/cell in the presence of 4% PEG 8000 (Merck Millipore) and 2% DMSO overnight.

Genotype D HBV was collected from the medium from HepAD38 cells without tetracycline. After concentration with Amicon Ultra centrifugal filter units (100 kDa), it was used for the attachment and internalization assay.

### 2.4. Reagents

Two drug libraries were purchased from Enzo Life Science Inc. (Ann Arbor, MI, USA): one is an FDA-approved drug library containing 760 compounds and the other is the Natural Product Library, Japanese version, containing 502 compounds. Skimmianine and myrcludex B were purchased from Selleck Biotech (Yokohama, Japan).

At the first screening, individual compounds were added at 2 μM for the FDA-approved drug library and at 4 μg/mL for the natural compound library. At the second screening, three concentrations of compounds from the FDA-approved drug library—0.2 μM, 2 μM and 20 μM—and the natural product library—0.04 μg/mL, 0.4 μg/mL and 4 μg/mL—were tested. Except for a few compounds that were eluted by water, most of the compounds were eluted by DMSO, and the final concentration of DMSO was adjusted to 2%.

### 2.5. HiBiT Assay

Extracellular HiBiT activity was determined by using the Nano Glo HiBiT Lytic Detection System (Promega, Madison, WI, USA), according to the manufacturer’s instructions, with the GloMax-Multi+Detection System (Promega). Forty microliters of medium were applied to the assay.

### 2.6. Quantitation of Extracellular HBV-DNA

Extracellular HBV-DNA levels were determined using an HBV Quantitative PCR (qPCR) Kit (Kubix, Hakusan, Japan) according to the manufacturer’s instructions.

### 2.7. Quantitation of HBsAg

HBsAg levels were measured using a HBs S Antigen Quantitative ELISA Kit, Rapid-II (Beacle, Inc., Kyoto, Japan) according to the manufacturer’s instructions.

### 2.8. Determination of the EC_50_

The amount of compound required to reduce HiBiT activity by 50% (EC_50_) was determined by Prism 9 (GraphPad Software, La Jolla, CA, USA).

### 2.9. Cell Viability Assay

Cell numbers were determined by a WST-8 assay using Cell Counting Kit-8 (DOJINDO, Kumamoto, Japan). The concentration of each compound required to reduce the cell number by 50% (CC_50_) was determined by Prism 9.

### 2.10. Live Cell Confocal Imaging

HepG2-NTCP-C4 cells were seeded at a density of 1.0 × 10^4^ cells/mL in chamber slides and incubated overnight at 37 °C in a 5% CO_2_ atmosphere. The next day, the medium was replaced with fresh growth medium containing Hoechst33342 for another 30 min. Then, the medium was replaced with fresh growth medium containing ReAsH-labeled TC155HBV particles. Time-lapse fluorescence images of ReAsH-labeled TC155HBV particles were acquired every 1 min for up to 120 min (120 frames) using an LSM510 inverted confocal microscope (Zeiss, Jena, Germany).

### 2.11. Attachment and Internalization Assay

The effect of skimmianine on attachment and internalization of HBV to hepatocytes was examined in [13,14]. In brief, HepG2-NTCP-YFP cells were pre-chilled on ice for 15 min and then incubated with 20,000 gEq/cell HBV derived from HepAD38 cells at 4 °C for 1.5 h with or without skimmianine. After incubation, for attachment assay, cells were vigorously washed 3 times to remove free HBV. For internalization assay, cells were cultured at 37 °C for 6 h, and they were then trypsinized and washed 3 times to remove cell-attached HBV. Each DNA sample was extracted and quantified by quantitative PCR. Heparin (Merck Millipore) was used as a negative control at a final concentration of 100 IU/mL.

### 2.12. Statistical Analysis

The results are expressed as the mean ± standard error of the mean. For the graphs, data were combined from at least three independent experiments. Statistical significance between two samples was examined by an unpaired *t*-test. The significance of between-group comparisons was tested by one-way analysis of variance with Bonferroni’s multiple comparisons test using Prism 9, and differences were considered statistically significant at a threshold of *p* < 0.05.

## 3. Results

### 3.1. Antiviral Screening of the Compound Library with Recombinant HBV

We developed a recombinant HBV in which a small peptide tag, HiBiT, was inserted into the N-terminus of preS1 of genotype C HBV [5]. This recombinant virus, HiBiT-HBV, is very useful for the high-throughput screening of anti-HBV compounds because the infection/replication of HiBiT-HBV in PXB cells can be monitored by measuring extracellular HiBiT activity. In other words, reduced extracellular activity with treatment with the tested compounds strongly implies that such compounds possess antiviral effects. In this study, we used HiBiT-HBV to test the 760 compounds in the FDA-approved drug library and the 502 compounds in the natural compound library. HiBiT-HBV from genotype C, namely HiBiT-HBV(Gt-C), was prepared, as reported previously [5], after the transfection of HepG2 cells with the HiBiT-HBV coding plasmid. Twelve hours after the infection of PXB cells with HiBiT-HBV(Gt-C), individual compounds were added at the single concentration. The medium was replaced with fresh medium containing fresh compound every 3 days, and on day 21 after infection, extracellular HiBiT activity was determined to test the antiviral effect, while a WST-8 assay was performed to assess cell viability. Then, HiBiT activity and absorbance from the WST-8 assay were normalized to those of the non-treatment group, to which DMSO or water was added. For the FDA-approved library, a well-known anti-HBV compound, entecavir (ETV), was included. In the ETV-treated cells, the relative HiBiT activity was 23% and the relative cell count from the WST-8 assay was 123%, which indicated that the antiviral effect and cell viability in this system were assessed as expected. As shown in Figure 1, most of the tested compounds exhibited neither an antiviral effect nor cell toxicity. However, several compounds showed an antiviral effect without cell toxicity, and we focused on those compounds in the second screening.

Then, we selected 96 compounds from the FDA-approved drug library and 76 compounds from the natural compound library for the second screening, in which we tested three concentrations of compounds from the FDA-approved drug library. PXB cells were infected with HiBiT-HBV(Gt-C), and 12 h after infection, the compounds tested were added at the above concentrations with the adjustment of the DMSO concentration to 2%. The medium was replaced with fresh medium containing fresh compound every 3 days, and on day 21 after infection, extracellular HiBiT activity was determined, and a WST-8 assay was performed. Then, HiBiT activity and absorbance from the WST-8 assay were normalized to those of the non-treatment group, to which only DMSO was added. For most tested compounds, dose-dependent antiviral effects were not observed or unexpected cytotoxicity was observed at lower concentrations. However, as shown in Figure 2, a total of 16 compounds exhibited dose-dependent antiviral effects without cytotoxicity. Of these, we focused on a natural product, skimmianine, because it showed a clear and strong dose-dependent antiviral effect without obvious cytotoxicity. Adapalen, which is categorized as retinoid, also seemed to be another candidate for further investigation because it showed a dose-dependent antiviral effect without obvious cytotoxicity. However, our group reported an inhibitory effect of retinoid on HBV replication by suppressing the sphingosine metabolic pathway [15]; thus, we did not further investigate adapalene.

### 3.2. Antiviral Effect of Skimmianine and Its Cytotoxicity

Skimmianine is a furoquinoline alkaloid present mainly in the Rutaceae family [16], with antioxidant, antimicrobial, and anti-inflammatory activities [17,18,19]. It also has a strong inhibitory effect on acetylcholinesterase [20]. During this research, to improve the fitness of HiBiT-HBV, we inserted the HiBiT coding sequence into the C-terminus of preS1 in genotype D HBV. This recombinant virus, designated HiBiT-HBV(Gt-D), showed more efficient infection and replication than that of genotype C. The N-terminus of the pre S1 protein of HBV is important for HBV entry, and it should be noted that the N-terminus of the preS1 protein of genotype D is 10–11 amino acids shorter than other genotypes including genotype C [21]. Then, we used HiBiT-HBV(Gt-D) for further study, and the detailed characterization of HiBiT-HBV(Gt-D) will be presented elsewhere. PXB cells were infected with the other recombinant virus, HiBiT-HBV(Gt-D), and 12 h after infection, skimmianine, DMSO, or ETV was added. The medium was replaced with fresh medium containing fresh compound and collected for HiBiT assay every 3 days, and on day 17 after infection, a WST-8 assay was performed to assess cell viability. In this time-course experiment using HiBiT-HBV(Gt-D), skimmianine suppressed extracellular HiBiT activity from the earlier time point in a dose-dependent manner (Figure 3a). However, cell viability sharply dropped between 1 μM and 10 μM (Figure 3b). The CC_50_ of skimmianine was calculated to be 3.27 μM, suggesting that the observed inhibitory effect of skimmianine at 10 μM and 50 μM was due to cytotoxicity, rather than an antiviral effect. Thus, we examined the possible antiviral effect of skimmianine at lower concentrations, where no cytotoxicity was expected to exist.

### 3.3. Antiviral Effect of Skimmianine at Lower Concentrations

PXB cells were infected with HiBiT-HBV(Gt-D), and 12 h after infection, skimmianine was added to the cells in a range from 100 pM to 10 μM. Then, the medium was replaced with fresh medium containing fresh compound and collected for HiBiT assay every 3 days until day 34. As expected, skimmianine at 5 μM and 10 μM strongly inhibited extracellular HiBiT activity, possibly due to its cytotoxicity. Surprisingly, skimmianine inhibited extracellular HiBiT activity, even at the lowest concentration, 100 pM, to about 50% compared with DMSO treatment on day 34 after infection (Figure 4a–c). When we calculated the EC_50_ on days 14, 20 and 34 after infection, it was 10.2 pM, 9.0 pM and 1.4 pM, respectively (Figure 4d), suggesting that the inhibitory effect of skimmianine was observed at early time points and did not change much during the infection.

### 3.4. Inhibitory Target of Skimmianine in HBV Infection

Although the above experiments suggested that skimmianine has an inhibitory effect on some steps of HBV infection, it was difficult to determine whether skimmianine inhibited the HBV entry step, which included attachment, internalization and retrograde trafficking, or affected the subsequent steps, such as cccDNA formation and HBV-DNA transcription, translation or amplification. To clarify the HBV lifecycle step inhibited by skimmianine, we added skimmianine at the same time as infection or 3 days after infection. In the former protocol, skimmianine would affect the entry step more strongly than the subsequent steps. On the other hand, in the latter protocol, skimmianine would affect the subsequent steps more strongly than the entry step. Thus, as shown in Figure 5a, we designated the former protocol “entry protocol” and the latter “replication protocol”. We also included myrcludex B, which is a well-known HBV entry inhibitor.

In the replication protocol, skimmianine did not have an inhibitory effect on HiBiT activity at 10 pM to 10 nM and began to have an inhibitory effect at more than 100 nM (Figure 5b,c). In the entry protocol, skimmianine had an inhibitory effect at 10 pM, which was the lowest concentration tested, and myrcludex B efficiently inhibited HiBiT activity, as expected (Figure 5b,c). While mircludex B at 200 nM almost completely blocked HBV entry, skimmianine at 10 pM-10 μM did not show such a strong inhibitory effect of mircludex B at 200 nM, indicating that mircludex B has a more potent inhibitory effect on HBV entry. When we calculated the EC_50_ using HiBiT activity for both protocols, it was 0.19 μM in the replication protocol and 0.36 pM in the entry protocol (Figure 5d). We also calculated the CC_50_ from the WST-8 assay for both protocols: it was 1.67 μM in the replication protocol and 1.87 μM in the entry protocol (Figure 5d). The selectivity index, calculated as the ratio of the CC_50_ to EC_50_, was 8.79 in the replication protocol and 5,100,000 in the entry protocol. These results indicate that skimmianine efficiently inhibits the entry step of HBV at low concentrations.

### 3.5. Effect of Skimmianine on HBV Replication in HepG2.2.15 Cells

To further investigate the effect of skimmianine on HBV replication, we used HepG2.2.15 cells, which stably support HBV replication. In HepG2.2.15 cells, cell-free reinfection of HBV released from the intracellular to extracellular environments is unlikely due to the very low expression level of the HBV receptor protein NTCP. Thus, we can specifically examine the effect of skimmianine on HBV replication. At 24 h after cell seeding, we treated HepG2.2.15 cells with skimmianine at 10 nM, 100 nM, 1 μM and 10 μM; entecavir at 200 nM; and DMSO at 2% and quantified the extracellular HBV-DNA level on days 7, 10 and 13. As shown in Figure 6a,b, while entecavir significantly reduced the extracellular HBV-DNA level, skimmianine did not cause a significant change at any concentration tested. When cell viability was assessed on day 13, neither entecavir nor any concentrations of skimmianine affected cell viability (Figure 6c). These results also suggest that skimmianine did not have an inhibitory effect on HBV replication.

### 3.6. Inhibition of Viral Entry by Skimmianine in Time-Lapse Fluorescence Imaging Analysis

To visualize HBV entry into the cell and its subsequent trafficking route in cells, we constructed HBV particles in which the capsid was tagged with tetracysteine (C-C-P-G-C-C) and labeled with the ReAsH fluorescent dye. The locations of the fluorescent signals for HBV core antigen (HBcAg) were examined with Hoechst33342 staining. The signal from the ReAsH fluorescent dye within the capsid could be visible after the virus enters the target cells, and the usefulness of this tetracyteine-tagged recombinant HBV for visualizing intracellular trafficking of HBV was reported [22]. HepG2-NTCP-C4 cells were infected with ReAsH-labeled HBV (ReAsH-TC155HBV, HBV derived from genotype C) that stably overexpressed the HBV receptor NTCP with or without skimmianine at 10 nM. Time-lapse fluorescence imaging analysis revealed that HBcAg started to accumulate in the nucleus at 60 min after infection and increased at 90 and 120 min after infection in the absence of skimmianine. On the other hand, skimmianine treatment abolished the nuclear accumulation of HBcAg (Figure 7 and Appendix A). This result once again indicated that skimmianine has an inhibitory effect on HBV entry.

### 3.7. Inhibitory Effect of Skimmianine on Viral Entry for Non-Recombinant HBV

In the above experiments, we used recombinant HBVs in which the HiBiT-coding sequence was inserted at preS1 and examined antiviral activity by measuring extracellular HiBiT activity. These recombinant HiBiT-HBV experiments raised several concerns: skimmianine may have an inhibitory effect on HiBiT activity itself, and skimmianine might inhibit only the entry step of the recombinant HiBiT-HBV and could be unable to inhibit the entry step of non-recombinant HBV without HiBiT insertion. To rule out these possibilities, we examined the effect of skimmianine at concentrations ranging from 10 pM to 100 nM on HBV infection by using non-recombinant HBV, which we designated WT-HBV. Genotype C HBV virus stock was purchased and then used to infect PXB cells. The effect of skimmianine was examined in both the entry and replication protocols, as shown in Figure 5a. The medium was replaced with fresh medium containing fresh skimmianine every 3 days, and then the amounts of HBsAg and HBV-DNA in the medium were quantitated by ELISA and quantitative PCR, respectively, on day 21 after infection. In the entry protocol, skimmianine statistically significantly reduced the HBsAg amount from 1 nM and the HBV-DNA amount from 10 pM. On the other hand, in the replication protocol, skimmianine did not significantly reduce the amounts of HBsAg and HBV-DNA at the tested concentrations (Figure 8). However, the suppressive effect of skimmianine on HBV entry for WT-HBV was not as potent as that for HiBiT-HBV. These results indicated that skimmianine inhibits HBV entry, not replication, although its antiviral effect on WT-HBV is weaker than that on HiBiT-HBV.

### 3.8. Effect of Skimmianine on Attachment and Internalization

Next, we further examined the effect of skimmianine on the attachment and internalization of HBV, both of which are an early step of HBV entry. HBV is known to attach to the cells at 4 °C independently from NTCP, and then to internalize to the cells dependent on NTCP at 37 °C [14]. We infected the genotype D HBV prepared from supernatants of HepAD38 cells to HepG2-NTCP-YFP cells or PXB cells with DMSO, several concentrations of skimmianine, or treatment with the attachment inhibitor heparin. Although heparin inhibited both attachment and subsequent internalization, as expected, skimmianine did not have any effect either on attachment or internalization both in HepG2-NTCP-YFP cells (Figure 9a) and PXB cells (Figure 9b). Taken together, these results suggest that skimmianine may inhibit translocation or retrograde trafficking to the endosomes/nucleus after uptake to the cells.

## 4. Discussion

This study demonstrates the usefulness of an infection system in which primary human hepatocytes, PXB cells, are infected with the recombinant virus, HiBiT-HBV, to identifying the antiviral compound from compound libraries, as we identified skimmianine. For this kind of drug screening, HepG2.2.15 cell lines, which stably support HBV replication, have been frequently used, as in a recent report [23]. In this case, to monitor the antiviral effect of each compound, the quantification of HBV-DNA or viral proteins, such as HBsAg, HBcrAg and HBeAg, is necessary from the first screening. However, the quantification of these viral markers is time-consuming and labor-intensive. Instead, when we use the recombinant virus HiBiT-HBV, we can monitor HBV infection and replication by simply measuring extracellular HiBiT activity, which is a much easier process. We infected PXB cells with HiBiT-HBV from the first screening. However, the availability of PXB cells is limited. Thus, it would be ideal to use hepatoma cell lines, such as Huh-7 or HepG2, stably overexpressing the HBV receptor NTCP, in addition to PXB cells because these hepatoma cells are more widely available than PXB cells. We reported that HiBiT-HBV showed lower infectivity than wild-type HBV without HiBiT insertion, which could hinder the monitoring of HBV infection and replication by extracellular HiBiT activity in HepG2-NTCP cells [5]. Further efforts to find a more suitable site for HiBiT insertion into the HBV genome without compromising viral fitness are needed to also use HiBiT-HBV in HepG2-NTCP and Huh-7-NTCP cells.

Following the first and second screenings using the HiBiT-HBVcc infection system in PXB cells, we identified skimmianine as a possible antiviral compound. Previously, Yand et al. examined the antiviral effects of several alkaloids from the roots of *Zanthoxylum nitidum*, including skimmianine [24]. In that report, they used HepG2.2.15 cells, which stably support HBV replication, and found that 0.2 μM skimmianine inhibited HBeAg secretion by around 31.9% without changing HBsAg secretion. These results indicate that skimmianine inhibits HBV replication, particularly transcription and/or translation, although it is not certain why skimmianine reduced only HBeAg. In the present study, we also examined the effect of skimmianine on HBV replication using HepG2.2.15 cells, but we did not observe any inhibitory effect of skimmianine on HBV replication when we measured HBV-DNA and HBsAg in the medium. Although we did not measure HBeAg or HBcAg, the effect of skimmianine on HBV replication seems very limited. In addition, when we added skimmianine to PXB cells 3 days after their infection with HiBiT-HBVcc, the inhibitory effect of skimmianine on HiBiT activity was not as strong as when we added it at the same time as the infection. These results also support a very weak antiviral effect of skimmianine on HBV replication.

The result of the imaging study by ReAsH-TC155HBV shown in Figure 7 suggests that skimmianine inhibits viral entry or the subsequent translocation or retrograde trafficking of the virus after internalization. The much stronger inhibitory antiviral effect of skimmianine observed in the “entry protocol” than in the “replication protocol” in Figure 5 also supports this idea. Furthermore, skimmianine did not inhibit either attachment or internalization, as shown in Figure 9. When all these findings are considered, skimmianine seems to inhibit the translocation or retrograde trafficking of the virus after entry. Compared with attachment and internalization, the mechanisms of retrograde trafficking remain unclear. Further study to clarify how skimmianine inhibits translocation or retrograde trafficking is needed in the future, as we recently reported that DOCK-11 plays an important role in retrograde trafficking [7]. While mircludex B at 200 nM almost completely blocks HBV entry, skimmianine at 10 pM-10 μM did not show such a strong inhibitory effect of mircludex B at 200 nM, as shown in Figure 5. This suggests that some viruses may escape from the inhibition of retrograde trafficking and initiate viable infections. Moreover, skimmianine is reported to have strong acetylcholinesterase (AChE) inhibitory activity [19]. Thus, it is expected to increase the level of acetylcholine. It remains unclear whether the observed inhibitory effect of skimmianine on HBV retrograde trafficking depends on AChE inhibitory activity. These kinds of hypotheses should be also examined in future work.

Because our work was performed using mainly PXB cells, which are primary human hepatocytes, and because PXB cells are more representative physiologically than hepatoma cell lines, such as HepG2 and Huh-7, the inhibition of HBV infection by skimmianine should be very reliable. Generally, alkaloids are expected to be used as antitumor medicines. Indeed, the CC_50_ of skimmianine for PXB cells was low, at around 1–4 μM, and it is challenging to advance to animal and/or human trials using skimmianine due to its high cytotoxicity. Furthermore, this study also highlighted a limitation of using HiBiT-HBV for antiviral screening. Although skimmianine exhibited a strong antiviral effect on HiBiT-HBV, it did not exhibit a similar antiviral effect on non-recombinant HBV, as shown in Figure 8. While it is difficult to pinpoint the specific reason why the antiviral effects of skimmianine on HiBiT-HBV and non-recombinant HBV were different, we speculate that the insertion of HiBiT into the preS1 region may cause this difference. The development of skimmianine derivatives, which show less cytotoxicity and the same or even more potent antiviral effect on non-recombinant HBV compared to skimmianine, is necessary to enable the testing of our findings in animal and/or human trials. This disparity in antiviral effect between HiBiT-HBV and non-recombinant HBV reminds us of the importance of validating the antiviral effect of the compounds identified using the recombinant virus in the non-recombinant virus.

In this study, we demonstrated the usefulness of the recently developed recombinant HiBiT-HBV in the high-throughput screening of HBV infection by using cell culture supernatants, as we identified skimmianine as a compound with an antiviral effect from compound libraries.

## Figures and Tables

**Figure 1 viruses-16-01346-f001:**
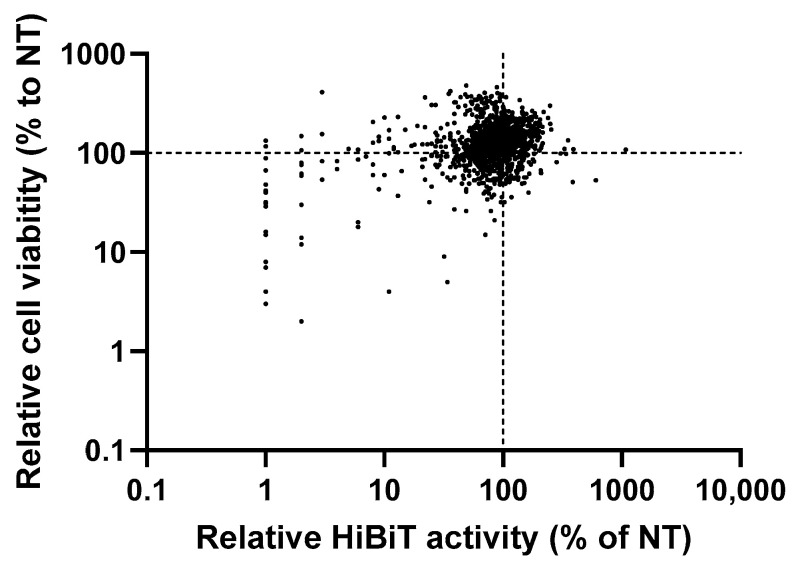
First screening of antiviral activity and cell toxicity using two kinds of compound libraries. Cell culture-derived HiBiT-HBV from genotype C, HiBiT-HBV(Gt-C)cc, was prepared as described in Section 2, and PXB cells were infected with HiBiT-HBV(Gt-C)cc at 1.5 × 10^8^ genome equivalent/cell. Twelve hours after infection, individual compounds were added at 2 μM for the FDA-approved drug library containing 760 compounds and at 4 μg/mL for the natural compound library containing 502 compounds. Except for a few compounds that were eluted by water, most of the compounds were eluted by DMSO, and the final concentration of DMSO was adjusted to 2%. The medium was replaced with fresh medium containing fresh compound every 3 days, and on day 21 after infection, extracellular HiBiT activity was determined to test the antiviral effect, while a WST-8 assay was performed to assess cell viability. Then, HiBiT activity and absorbance from the WST-8 assay were normalized to those of the non-treatment group, to which DMSO or water was added. Each compound was tested in a single well of a 96-well plate. The relative ratio to the non-treatment of each compound is shown as a dot, and each dot shows both the relative HiBiT activity and relative cell viability of individual compounds. The dashed lines indicate the measures from the non-treatment (NT) cells.

**Figure 2 viruses-16-01346-f002:**
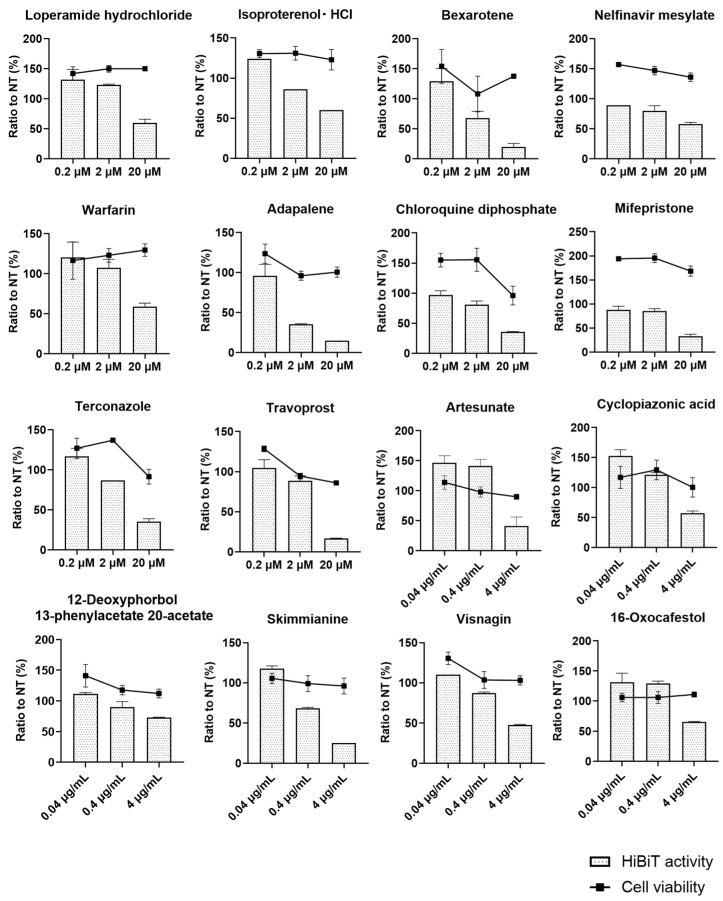
Second screening of antiviral activity and cell toxicity at serially diluted concentrations. Ninety-six compounds from the FDA-approved drug library and 76 compounds from the natural compound library were selected for the second screening, in which we tested three serially diluted concentrations of compounds from the FDA-approved drug library—0.2 μM, 2 μM and 20 μM—and the natural product library—0.04 μg/mL, 0.4 μg/mL and 4 μg/mL. PXB cells were infected with HiBiT-HBV(Gt-C) at 1.5 × 10^8^ genome equivalent/cell, and 12 h after infection, the compounds were added at the above concentrations with adjustment of the DMSO concentration to 2%. The medium was replaced with fresh medium containing fresh compound every 3 days, and on day 21 after infection, extracellular HiBiT activity was determined and a WST-8 assay was performed. Then, HiBiT activity and absorbance from the WST-8 assay were normalized to those of the non-treatment group, to which only DMSO was added. Each compound was tested using three wells of a 96-well plate. The bar graph shows the relative HiBiT activity, the line graph shows the relative cell viability, and the error bar shows the standard deviation from the measurements of three wells. Here, we show only the results of the compounds that inhibited HiBiT activity in a dose-dependent manner without greatly affecting cell viability.

**Figure 3 viruses-16-01346-f003:**
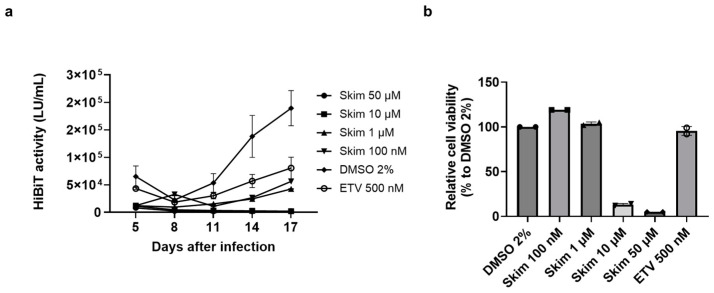
Time-course effect of skimmianine on HBV infection and its effect on cell viability. Cell culture-derived HiBiT-HBV from genotype D, HiBiT-HBV(Gt-D)cc, was prepared as described in Section 2. PXB cells were infected with HiBiT-HBV(Gt-D)cc, and 12 h after infection, skimmianine was added at 100 nM, 1 μM, 10 μM and 50 μM, along with ETV at 500 nM and DMSO at 2%. The medium was replaced with fresh medium containing fresh compound and collected for HiBiT assay every 3 days until day 17. Then, a WST-8 assay and microscopic analysis were performed to assess cell viability. Each condition was tested using three wells of a 96-well plate. (**a**) The extracellular activity at each condition and time point is shown, and the error bar shows the standard deviation from the measurements of the three wells. (**b**) Cell viability relative to that of the DMSO control on day 17 at each condition, and the error bar shows the standard deviation from the measurements of three wells. Each symbol shows the individual measurements. Skim, skimmianine; ETV, entecavir.

**Figure 4 viruses-16-01346-f004:**
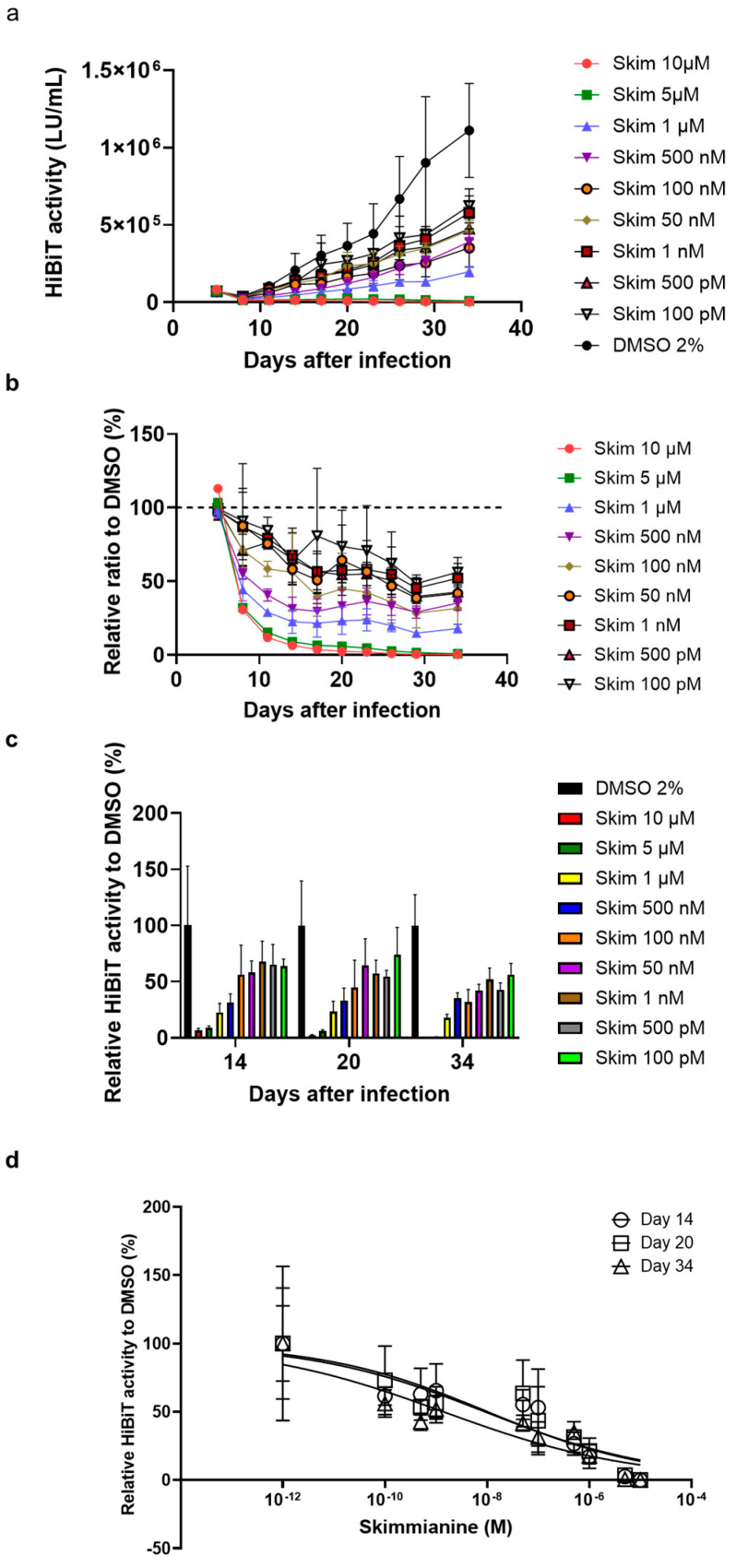
Effect of skimmianine on HBV infection at lower concentrations. PXB cells were infected with HiBiT-HBV(Gt-D)cc, and12 h after infection, skimmianine was added to the cells in a range from 100 pM to 10 μM. Then, the medium was replaced with fresh medium containing fresh compound and collected for HiBiT assay every 3 days until day 34. Each condition was tested using three wells of a 96-well plate. The error bar shows the standard deviation from the measurements of three wells. (**a**) HiBiT activity at each condition and time point. (**b**) Relative HiBiT activity to that of the DMSO control at each condition and time point. HiBiT activity of the DMSO control at each time point was set to 100% and is shown with a dashed line. (**c**) Relative HiBiT activity to that of DMSO control on days 14, 20 and 34 at each condition. (**d**) The dose–response curve was plotted using HiBiT activity relative to that of DMSO control on days 14, 20 and 34 to calculate the EC_50_ on each day.

**Figure 5 viruses-16-01346-f005:**
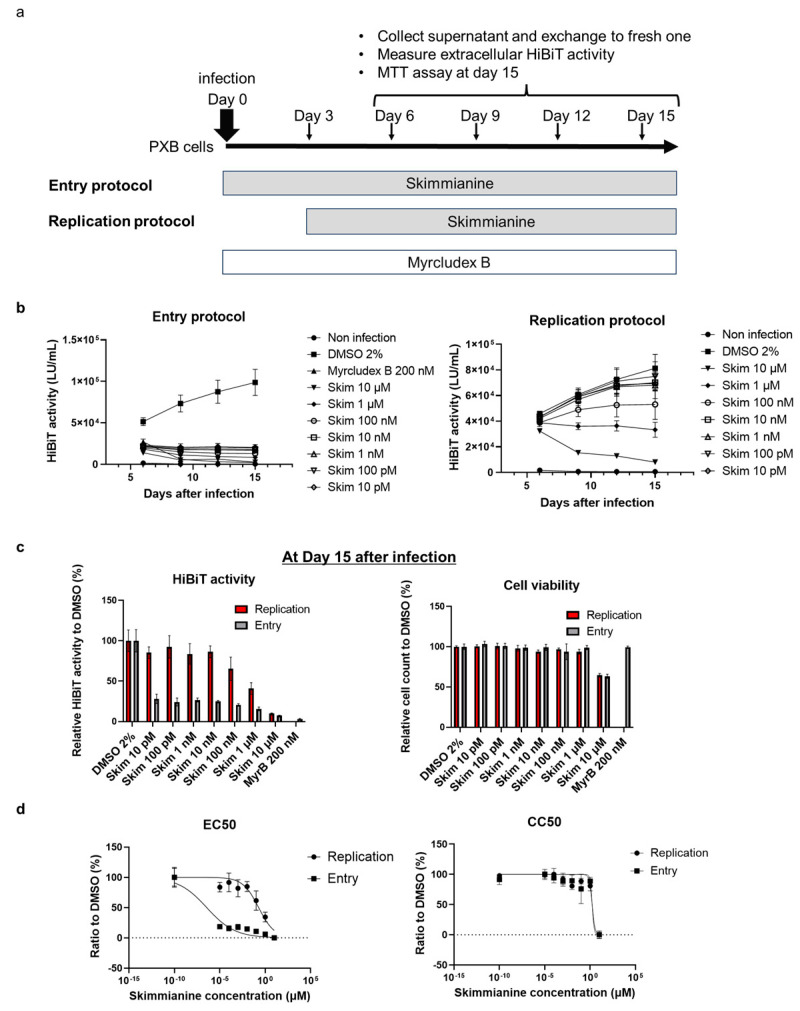
Inhibitory target of skimmianine in HBV infection. PXB cells were infected with HiBiT-HBV(Gt-D)cc, and to clarify the inhibitory target of skimmianine within the HBV lifecycle, skimmianine was added at 10 pM to 10 μM at different time points: at the same time as infection or 3 days after infection. The former protocol was designated the “entry protocol”, while the latter was designated the “replication protocol”. Myrcludex B, which is a well-known HBV entry inhibitor, was also added at the same time as infection at 200 nM. Then, the medium was replaced with fresh medium containing fresh compound and collected for a HiBiT assay every 3 days until day 15. Each condition was tested using three wells of a 96-well plate. The error bar shows the standard deviation from the measurements of three wells. On day 15, a WST-8 assay was also performed to examine cell viability. (**a**) The protocol of this analysis. (**b**) HiBiT activity at each condition and time point for each protocol. (**c**) Relative HiBiT activity to that of DMSO control for each protocol on day 15 after infection. The left panel shows the HiBiT activity results, while the right panel shows the cell viability results. (**d**) The dose–response curve was plotted using relative HiBiT activity (left panel) and cell viability (right panel) to those of DMSO control on day 15 after infection to calculate the EC_50_ and CC_50_, respectively. The dashed line indicates the 0% ratio to the DMSO control. Skim, skimmianine; MyrB, myrcludex B.

**Figure 6 viruses-16-01346-f006:**
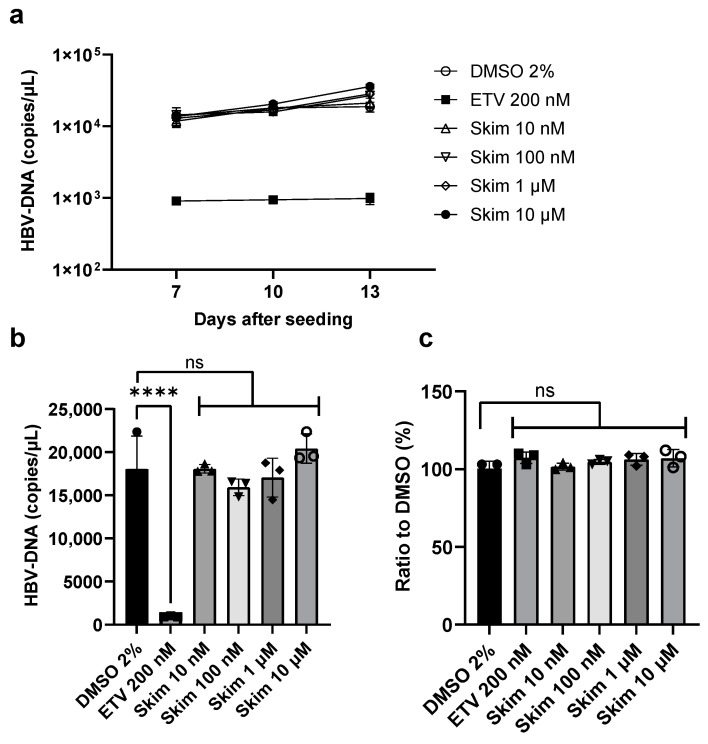
Effect of skimmianine on HBV replication in HepG2.2.15 cells. HepG2.2.15 cells were seeded in 96-well plates, and 24 h later, skimmianine was added at 10 nM, 100 nM, 1 μM and 10 μM; entecavir at 200 nM; and DMSO at 2%. The medium was then replaced with fresh medium containing fresh compound and collected for the quantitation of the HBV-DNA level. On days 7, 10 and 13, the extracellular HBV-DNA level was quantitated by quantitative PCR and cell viability was analyzed on day 13 by a WST-8 assay. Each condition was tested using three wells of a 96-well plate. (**a**) HBV-DNA level in each condition. (**b**) HBV-DNA levels on day 10. Error bars show the standard deviation from the three wells. Each symbol shows the individual measurements. The differences in the means between 2% DMSO and each condition were analyzed with one-way analysis of variance with Bonferroni’s multiple comparisons test. **** *p* < 0.0001. (**c**) Cell viability relative to 2% DMSO. Each symbol shows the measurements. The differences in the means between 2% DMSO and each condition were analyzed with one-way analysis of variance with Bonferroni’s multiple comparisons test. ETV, entecavir; Skim, skimmianine; ns, not significant.

**Figure 7 viruses-16-01346-f007:**
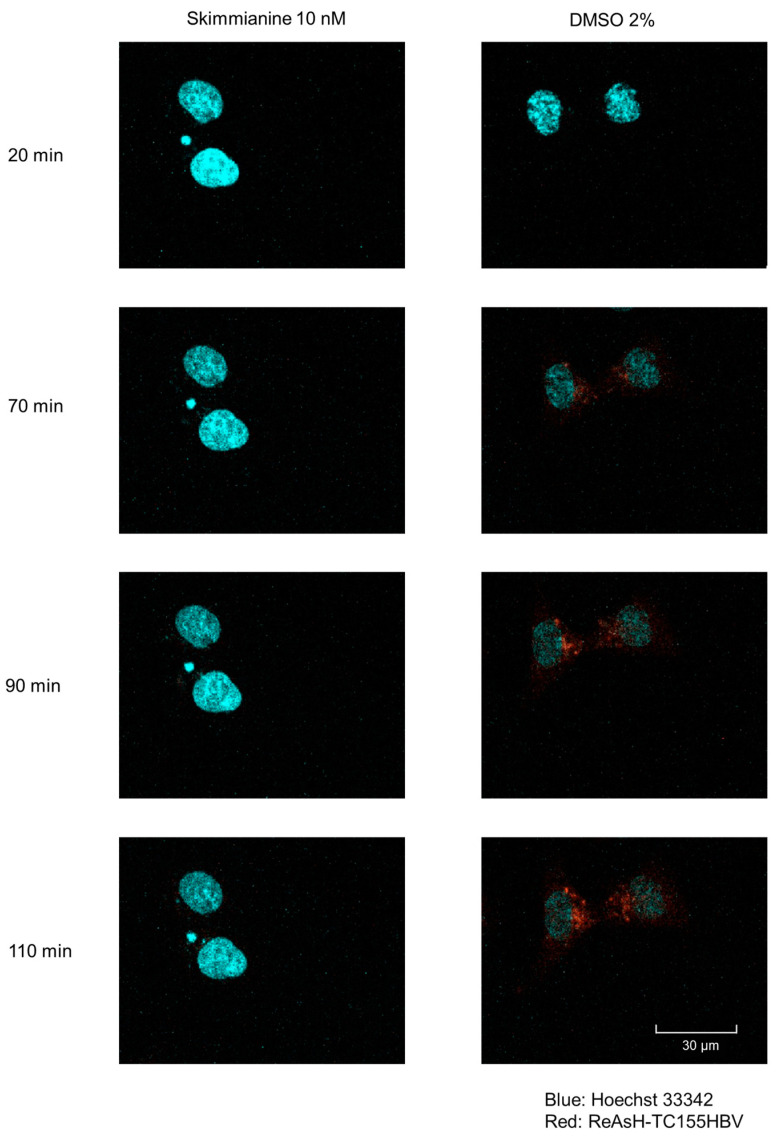
Inhibition of viral entry by skimmianine in time-lapse fluorescence imaging analysis. HepG2-NTCP-C4 cells were infected with ReAsH-labeled HBV (ReAsH-TC155HBV; genotype C HBV) that stably overexpressed the HBV receptor NTCP with or without skimmianine at 10 nM. Time-lapse fluorescence images of ReAsH-labeled TC155HBV particles were acquired every 1 min for up to 120 min. The images shown were obtained at 20, 70, 90 and 110 min after infection. Blue: Hoechst 33342, Red: ReAsh-TC155HBV.

**Figure 8 viruses-16-01346-f008:**
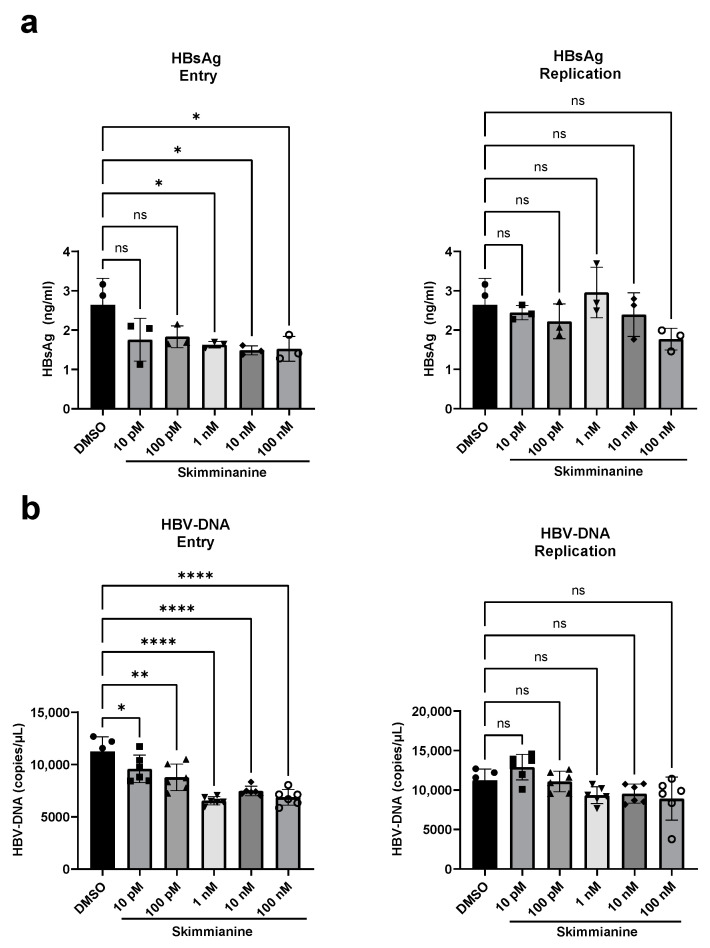
Inhibitory effect of skimmianine on viral entry for non-recombinant HBV. Non-recombinant HBV from genotype C, designated WT-HBV, to which the HiBiT coding sequence was not inserted, was used to infect PXB cells. Then, skimmianine at concentrations ranging from 10 pM to 100 nM was added in both the entry and replication protocols, as shown in Figure 5a. The medium was replaced with fresh medium containing fresh skimmianine every 3 days, and then the amounts of HBsAg (**a**) and HBV-DNA (**b**) in the medium were quantitated by ELISA and quantitative PCR, respectively, on day 21 after infection. Each symbol shows the individual measurements. Differences in the means between DMSO treatment and each treatment were analyzed via a one-way analysis of variance test. * *p* < 0.05, ** *p* < 0.01, and **** *p* < 0.0001. ns, not significant.

**Figure 9 viruses-16-01346-f009:**
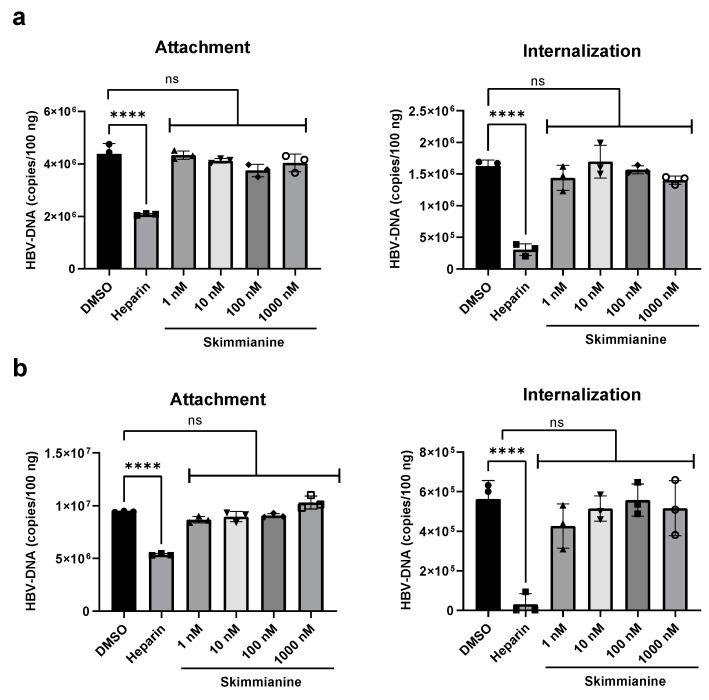
Effect of skimmianine on attachment and internalization. HepG2-NTCP-YFP cells or PXB cells were pre-chilled on ice for 15 min and then incubated with 20,000 gEq/cell HBV derived from HepAD38 cells at 4 °C for 1.5 h either with indicated concentrations of skimmianine or without skimmianine (DMSO 2%). Heparin was used as a negative control at a final concentration of 100 IU/mL. After incubation, for attachment assay, cells were vigorously washed 3 times to remove free HBV. For internalization assay, cells were cultured at 37 °C for 6 h, and then they were trypsinized and washed 3 times to remove cell-attached HBV. Each DNA sample was extracted and quantified by quantitative PCR and normalized to 100 ng total DNA. (**a**) The results from HepG2-NTCP-YFP. (**b**) Those from PXB cells. Each symbol shows the individual measurements. Differences in the means between DMSO treatment and each treatment were analyzed via a one-way analysis of variance test. **** *p* < 0.0001. ns, not significant.

## Data Availability

Data are contained within the article.

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
