# Peer review of "High-Throughput Screening of Antiviral Compounds Using a Recombinant Hepatitis B Virus and Identification of a Possible Infection Inhibitor, Skimmianine"

_viruses, 2024, doi:10.3390/v16081346_

Round 1

Reviewer 1 Report (Previous Reviewer 1)

Comments and Suggestions for Authors

The authors addressed satisfactorely the concerns of the reviewers.

Author Response

The authors addressed satisfactorily the concerns of the reviewers.

Answer: Thank you so much for reviewing the revised manuscript.

Reviewer 2 Report (New Reviewer)

Comments and Suggestions for Authors

I carefully reviewed the article “High-throughput Screening of Antiviral Compounds Using a Recombinant Hepatitis B Virus and Identification of a Potent Infection Inhibitor, Skimmianine” by Mika Yoshita, et., and I think that the present revised version of the article is well organized, the novelty of the results is very good, the research design and methodology is appropriate, and the presentation of the paper is suitable Viruses journal.

The current approved therapy in HBV infection is based on antivirals that selectively inhibit the viral reversetranscriptase and the viral entry, but cannot determine viral clearance. Skimmianine can represent a non-toxic approach for potential clinical applications in inhibiting HBV replication, so I consider the present study is novel and significant for the clinical practice. Moreover, as HBV uses the retrograde trafficking route that can serve to maintain cccDNA, Skimmianine can be used to prevent persistent HBV infection. 

My suggestion is related to the toxicity of the reported compound – maybe a short statement about the non-toxicity of Skimmianine can be added in the Conclusion section.

The utility of the recombinant HiBiT-HBV in high-throughput screening of HBV infection is also valuable for future advances in antiviral therapy.

Comments on the Quality of English Language

Minor editing of English language is required – please review the text: i.e. in the abstract – line 40 “demonstrates the usefulness”, etc.

Author Response

My suggestion is related to the toxicity of the reported compound – maybe a short statement about the non-toxicity of Skimmianine can be added in the Conclusion section.

Answer:

  • We corrected the sentence at the discussion (line 516-517) as following.

‘This indicates that skimmianine does not show cytotoxicity at the concentrations where the strong inhibitory effect on retrograde trafficking of HBV is observed.’

  • We also add the statement about cytotoxicity of skimmianine in conclusion (line 520) as following.

‘in which we identified skimmianine as a new antiviral against HBV without cytotoxicity,

Minor editing of English language is required – please review the text: i.e. in the abstract – line 40 “demonstrates the usefulness”, etc.

Answer: Corrected

This manuscript is a resubmission of an earlier submission. The following is a list of the peer review reports and author responses from that submission.

Round 1

Reviewer 1 Report

Comments and Suggestions for Authors

The authors used a recombinant HBV virus, with and HiBit insertion as a high throughput system for monitoring antiviral activity. They tested more than 1200 compounds and found The information is important. Some concerns should be addressed before accepting the manuscript.

1.   Figure 1: There is too much information in the legend of the methodology and in contrast, the meaning of each axis is not explained.

2.   The same observation applies to Results in page 4: too much information which should be moved to methodology, and few description of the results. The actual description of results is in lines 175-181.

3.   Same observation for lines 204-208.

4.   Figure 2. Same observation than Figure 1. In addition, the size of the letter in the axis is small.

5.   Figure 2. The authors focused on skimmianine. Why not on adapalene, which exhibited even better selectivity index?

6.   Same observation for lines 240-243 that point 2.

7.   Figure 5. Same observation than Figure 1. The letters are too small.

8.   Why mircludex was not tested in the replication protocol, as a negative control?

9.   Skimmianine seems to inhibit HBV entry, but at a higher concentration compared to mircludex. This was not stated in Results.

10.   Figure 8: were the differences significant? It would be interesting to include mircludex as a control in these experiments.

11.   How many replicas of the experiments were done?

12.   The authors did not test for virucidal activity of skimmianine.

13.   For testing inhibition of entry, it would be interesting to analyze the effect of skimmianine just preincubating the cells before infection, maintaining it during the entry process, and then eliminate the drug and the recombinant viruses which did not enter by repeated washes, using mircludex as a control.

14.   The authors indeed show a very performant high throughput system for monitoring antiviral activity. Their results suggest (the authors should be more cautious and not state ¨clearly showed¨ (see for example line 435.) that skimmianine may inhibit HBV entry. However, their results with real HBV virions were not as convincing as the ones with their system. Perhaps some viruses may escape form the entry inhibition and initiate viable infections. The authors should include this in limitations of this study.

Reviewer 2 Report

Comments and Suggestions for Authors

In their manuscript  entitled “High-throughput Screening of Antiviral Compounds Using a  Recombinant Hepatitis B Virus and Identification of a Potent Entry Inhibitor, Skimmianine” Mika Yoshita and coworker describe the use of a recombinant  tagged HBV virus for the screening of antiviral compounds. Based on this system they identify 16 compounds with an antiviral effect. Based on this initial screening, they focus on a more detailed analysis of Skimmianine as a potential antiviral. Based on their experiments they conclude that Skimmianine primarily acts as an entry inhibitor and has no significant effect on the post entry steps as genome expression, genome replication, virus morphogenesis and release.

This could be an interesting observation but the design of the study is not fully convincing.

If the authors claim that Skimmianine acts an an entry inhibtor classic analysis of the effect on virus entry must be performed. They must investigate if the attachment, the binding or the entry is affected. For the entry analysis they should quantify the impact on the number of internalized genomes by cell fractionation experiments followed by qPCR.

Moreover, the authors should provide data which allow to discriminate between the effect on formation/release of subviral particles and viral particles. In light of the fact that a huge excess of subviral particles is formed as compared to viral particles an interference with the formation of viral particles can be underestimated. In this context the use of HepG2.2.15 cells (they do not replicate HBV, the express HBV from an integrate) is critical as they overproduce a lot of subviral particles as compared to virions. 

The authors should consider to do some experiments with HBV gtG, which fails to release significant amount of HBsAg.

Some  data based on the elegant recombinant virus system should be corroborated by quantification of the number of released genomes and/or by determination of the TCID50.

The staining for the internalization of the tetracys-tagged virus is astonishing. There are only very few virus particles internalized and capsids transported towards the nucleus. The authors should comment on the strong signals they observe (fig. 7).

As compared to the other genotypes the N-terminus of the PreS1 domain strongly differs  from the other genotypes du to the lack of 10-11 aa. This should be mentioned when the gtD-based construct is introduced.

The study is almost completely descriptive. There are no mechanistic data provided characterizing the inhibitory effect of this substance on HBV life cycle.

Comments on the Quality of English Language

moderate changes are required